# Effects of Oxide Fragments on Microstructure and Mechanical Properties of AA6061 Aluminum Alloy Tube Fabricated by Thermomechanical Consolidation of Machining Chips

**DOI:** 10.3390/ma16041384

**Published:** 2023-02-07

**Authors:** Zhen Zhang, Jiamiao Liang, Tian Xia, Yuehuang Xie, Sammy Lap Ip Chan, Jun Wang, Deliang Zhang

**Affiliations:** 1Shanghai Key Laboratory of Advanced High-Temperature Materials and Precision Forming and State Key Laboratory of Metal Matrix Composites, School of Materials Science and Engineering, Shanghai Jiao Tong University, Shanghai 200240, China; 2Changzhou Reborn Advanced Alloy Materials Technology Co., Ltd., Changzhou 213166, China; 3Jiuhe Frontier Innovations of Science and Technology (Shanghai) Co., Ltd., Shanghai 201100, China; 4Department of Chemical and Materials Engineering, National Central University, Zhongli 320317, Taiwan; 5School of Materials Science and Engineering, University of New South Wales, Sydney, NSW 2052, Australia; 6State Key Laboratory of Rolling and Automation, Northeastern University, Shenyang 110819, China

**Keywords:** solid-state recycling, recrystallization, inter-chip bonding

## Abstract

An AA6061 aluminum alloy tube was fabricated by compacting machining chips via thermomechanical consolidation, including hot pressing and hot extrusion. The microstructure evolution and formation of oxide particles were investigated in correlation to tensile mechanical properties. It was found that fine Al/Mg oxide particles were formed due to the fracture of oxide layers on the chips and the reaction between Mg and Al_2_O_3_ during hot extrusion. The oxide particles inhibited the growth of recrystallized α-Al grains, leading to the formation of a microstructure consisting of coarse elongated grains with sizes of 420 μm and fine equiaxed grains with sizes of 10 μm. After T6 heat treatment, a microstructure with finer grains (grain sizes: 34 μm) formed due to further recrystallization induced by residual strain. The tensile mechanical properties testing results indicated that a good combination of strength (296 MPa) and ductility (7.6%) was achieved in the T6 heat treated samples, which were likely attributed to the high-quality inter-chip bonding, as well as the fine oxide particles which were small enough that further crack nucleation and growth around them were inhibited during tensile deformation. For the purpose of comparison, the microstructure and mechanical properties of the as-extruded and T6 heat treated samples produced by hot extrusion of the cast ingot of AA6061 aluminum alloy were also investigated. The lower tensile strength of solid-state recycled tube sample might be associated with the consumption of Mg atoms by the oxidation reaction, leading to the lower density of β″ precipitates in precipitation strengthening.

## 1. Introduction

Aluminum alloys have been applied widely in the fields of aviation, aerospace, architecture, automotive, and other structural components due to their high specific strength and good corrosion resistance. The considerable energy consumption of primary aluminum production makes the recycling of aluminum alloy chips produced from machining and turning to be a concern [1,2,3,4]. In comparison with the conventional remelting process, solid-state recycling, which converts the chips into consolidated materials directly without melting, is more efficient in terms of energy consumption and material utilization as well as environmentally-friendly [3,5,6]. The rationale behind solid state recycling technique is based on the severe plastic deformation, introducing large shear deformation to enhance physical and metallurgical bonding. To date, an increasing number of solid-state recycling processes have been reported including ball milling, sintering [7], hot pressing [8,9,10], spark plasma sintering [11], hot extrusion [3,5,6,12,13,14,15,16,17,18,19,20,21,22,23,24], equal channel angular pressing [25,26,27,28,29], cycling extrusion compression [30,31], friction stir extrusion [32,33], high-pressure torsion [34], and screw extrusion [35].

It is well recognized that the plastic deformation introduced during solid-state recycling could induce the fracture of oxide films of the aluminum chips and dynamic recrystallization of α-Al grains. The extent of the fracture of oxide films and dynamic recrystallization is closely related to deformation temperature, strain rate, and extrusion ratio. Hasse et al. [25] found that plastic deformation at an elevated temperature caused the breaking of oxide layers into fine particles, producing a large area of atomically fresh surfaces on the aluminum chips, which could enhance the atomic diffusion across the inter-chip boundaries. Peng et al. [36] reported that the inter-chip boundaries almost disappeared with increasing the extrusion ratio to 20:1 at 400 °C, and they stated that defects such as grain boundaries, particle surfaces, and dislocations could act as fast diffusion paths. Yang et al. [8] stated that sufficient bonding was established between individual swarf pieces of Ti-6Al-4V during the short hot pressing. Liang et al. [17] found that recrystallization and grain growth of the solid-state recycled Al–7Si–0.3Mg alloys were induced by plastic deformation during hot extrusion, which was also accompanied by a rapid establishment of strong bonding between individual chips. Hasse et al. [25] reported that severe plastic deformation during integrated equal channel angular pressing (iECAP) contributed to a higher combination of tensile strength and ductility of the solid-state recycled aluminium alloy materials. They attributed it to the finer equiaxed grains and higher inter-chip bonding strength, as more effective in inducing dynamic recrystallization and breaking Al_2_O_3_ surface films into fine particles.

The mechanical properties of recycled aluminium alloy materials are closely related to their microstructures. Luo et al. [27] found that the fragments of oxide films could cause additional dispersion strengthening to the titanium samples fabricated by hot extrusion of titanium chips. Chino et al. [13] believed that the breaking of oxide layers of aluminum alloy chips were responsible for the lower elongation to fracture of the specimens fabricated by hot extrusion of the aluminum alloy chips, as compared to that of the samples fabricated by hot extrusion of the cast ingot. Suzuki et al. [37] reduced the defects at the inter-chip boundaries by differential speed rolling, and consequently, the mechanical properties and corrosion resistance of chips based on 6061 aluminum alloy were improved.

As highlighted above, large amounts of previous works have shown that high-quality inter-chip bonding could be achieved by introducing plastic deformation during the consolidation of aluminium alloy chips, which facilitated the fragments of oxide layers on the chips and enhanced the atomic diffusion across the inter-chip boundaries. However, to the best of the author’s knowledge, there is still a lack of investigation about the effects of oxide fragments on microstructure evolution and mechanical properties of solid-state recycled aluminum alloys. In this study, we primarily investigated the effects of oxide fragments on microstructure features and correlated them to the tensile mechanical properties and fracture behaviour of a tube fabricated by hot extrusion of AA6061 chips compact. Moreover, the tensile properties in both longitudinal and transverse directions influenced by the anisotropic microstructures have been studied as well. For comparison, a tube fabricated by hot extrusion of AA 6061 cast ingot was also investigated. This study establishes a deep understanding of good inter-chip bonding and anisotropic microstructure influenced by oxide layers/particles.

## 2. Experimental Procedure

The raw materials used in this study were AA6061 aluminum alloy machining chips. As shown in Figure 1a, the chips had a length of 3–10 mm, a width of 1–2 mm, and a thickness of 0.1–0.5 mm. The tap density of the chips was 0.33 g/cm^3^. The AA6061 chips were firstly washed in soap water to remove the residual lubricants, then they were dried at 55 °C for 5 h. The dried chips were dropped into a cylindrical H13 steel die and then heated by a ceramic heater band up to 400 °C (holding time: 10 min) under the protection of argon, followed by hot pressing (200 ton) for 5 min. The compact was then extruded at 480 °C with an extrusion ratio of 58:1 to produce a tube with an inner diameter of 15 mm and an outer diameter of 19 mm. After extrusion, the tube was quickly transferred into water to obtain the microstructure of extruded state. For the purpose of comparison, an ingot metallurgy tube produced by hot extrusion of AA6061 cast ingot was used as a counterpart. The as-extruded tubes were heat treated with standard T6 heat treatment procedures: solution treatment at 535 °C for 2 h, quenching into water, and then artificial aging at 177 °C for 8 h. The details of the fabrication processes and the corresponding sample notations are listed in Table 1.

The chemical compositions of the tube samples were determined by an inductively coupled plasma (ICP) emission spectrometer (iCAP7600, Thermo, Waltham, MA, USA). The oxygen contents were measured by a LECO TCH-600 nitrogen/oxygen/hydrogen analyzer, LECO, St. Joseph, IL, USA), which were 2.52 wt.% for the SSR tube sample and 0.45 wt.% for the IM tube sample. The microstructures of the tube samples were characterized by scanning electron microscopy (SEM) (Nova NanoSEM 230, FEI, Valley City, ND, USA), electron backscatter diffraction (EBSD) (Aztec HKL Max, Oxford Instruments, Oxford, UK) on SEM, and transmission electron microscopy (TEM) (JEM-2100F, JEOL, Tokyo, Japan). The EBSD specimens were firstly cut along the longitudinal direction (extrusion direction) of the tube samples and then were processed by grounding, mechanically polishing, and electrolytically polishing using an electrolyte of 90% methanol + 10% perchloric acid with a parameter voltage of 25 V for 10 s. The TEM foils were prepared by twin-jet electrochemical thinning using an electrolyte of 70% methanol, + 30% nitric acid, with a voltage of 30 V at −20 °C.

For the investigation of anisotropy of tensile mechanical properties, specimens along the longitudinal and transversal directions of the tube samples were prepared. Figure 1b shows schematically how the longitudinal and transversal tensile test specimens were cut from the tube samples. The longitudinal tensile test specimens of the tube samples were cut into dog-bone shaped specimens with a gauge length of 15 mm and cross-section dimensions of 2.0 × 1.5 mm^2^ by wire-electrode cutting and then mechanically grounded to 2000 mesh abrasive paper. For preparing the transversal tensile test specimens, the tube samples were cut into two halves and then flattened into plates using a hydraulic press. The transversal tensile test specimens were dog-bone shaped with a gauge length of 10 mm and a cross-section area of 1.6 × 1.4 mm^2^. The tensile tests were performed using a Zwick/Roell Z100 testing machine (Zwick, Ulm-Einsingen, Germany) with a strain rate of 5 × 10^−4^ s^−1^. Extensometer was used to measure the elongation to fracture. The tensile test was performed based on GB/T 228-2002 standard (excluding specimen size due to the limited dimension of samples, the grade of load cell and extensometer are Grade 1). Three specimens were tested to ensure consistency.

## 3. Results

Figure 2 shows the EBSD images of the as-extruded and T6 heat treated samples, and the corresponding grain size distributions and misorientation angle distributions. As shown in Figure 2a, the SSR sample demonstrated a microstructure consisting of coarse elongated grains with length and width in the range of 200–800 μm and 150–450 μm, respectively, and fine equiaxed grains with sizes in the range of 2–20 μm. The corresponding grain size distribution result showed that the average grain size was 60 μm. As shown in Figure 2b, the IM sample exhibited a homogenous microstructure consisting of equiaxed grains with an average grain size of 35 μm. After T6 heat treatment, a microstructure containing both elongated and equiaxed grains was observed in the SSR(L)-T6 sample, as shown in Figure 2c. Based on the grain size distribution, the average grain size of the SSR(L)-T6 sample was 34 μm, showing an apparent decrease in grain size as a result of the heat treatment. For the IM sample, after T6 heat treatment, the microstructure consisted of equiaxed Al grains with an average size of 60 μm, as shown in Figure 2d. This indicated an apparent increase in grain size due to grain growth during the T6 heat treatment. As shown in Figure 2e, the microstructure containing both elongated and equiaxed grains was observed in the SSR(T)-T6 sample; its corresponding average grain size was 42 μm. As shown in Figure 2f, the IM(T)-T6 sample also showed a microstructure consisting of equiaxed Al grains which is similar to those of the IM and IM(L)-T6 samples; its average grain size was 43 μm.

According to the grain misorientation angle distribution of the as-extruded and T6 heat treated tube samples, it can be found that the fractions of high angle grain boundary (HAGB), low angle grain boundary (LAGB) and sub-grain boundary (SGB) were 53%, 5%, 42%, and 76%, 10%, and 14%, respectively, in the SSR and IM samples (Figure 2a,b). After T6 heat treatment, for the SSR(L)-T6 sample, the fractions of HAGB and LAGB increased to 80% and 12% respectively, while the fraction of SGB decreased to 8%, as shown in Figure 2c. In contrast, the fraction of HAGB increased to 84%, and the fractions of LAGB and HGB decreased to 9% and 7%, respectively, in the IM(L)-T6 sample (Figure 2d). For the SSR(T)-T6 and IM(T)-T6 samples, the fractions of HAGB, LAGB, and SGB were 93%, 2%, 5%, and 91%, 3%, and 6%, respectively (Figure 2e,f).

As shown in Figure 3, the SSR sample had a mixture of <1 0 0> recrystallization texture and <1 1 1> deformation texture, while the IM sample had only a <1 0 0> recrystallization texture in the microstructure. Similar <1 0 0> recrystallization textures were also observed in the IM(L)-T6 samples. However, the SSR(L)-T6 sample changed a lot in that <100> fiber texture was much stronger, and <111> fiber texture even disappeared. The inverse pole figures results confirm that large amounts of deformed structures in the SSR sample remained after hot extrusion and they were restored in SSR-T6 via recrystallization.

As shown in Figure 4a, fine oxide particles were observed in the fine grain zones of the microstructure of the SSR sample. These oxide particles were mainly distributed along the grain boundaries. The corresponding EDS element mapping indicated that these particles were rich in Al, Mg, and O elements (Figure 4b). It is speculated that the particle is likely MgAl_2_O_4_, which will be mentioned in the discussion section. Based on the particle size distribution shown in Figure 4c, it was determined that the average size of oxide particles was 1.94 μm. Figure 4d illustrated the strain contouring around the fine oxide particles shown in Figure 4a, it was clear that the magnitude of residual strain was high in the vicinity of these oxide particles.

Figure 5 shows the TEM bright field images of the SSR(L)-T6 and IM(L)-T6 samples. The inserts are the corresponding selected-area electron diffraction (SAED) patterns acquired along the Al [0 0 1] zone axis direction. It can be seen that a high density of needle-shaped precipitates, with a diameter of 4 nm and length of 10–40 nm, were observed inside the grains. These precipitates are β phases which are usually observed in 6xxx series aluminum alloys at the peak aging stage and are responsible for precipitation strengthening of the material. It was also found that the amount of the β″ precipitates in the SSR(L)-T6 sample was slightly lower than that in the IM(L)-T6 sample.

Figure 6 shows the tensile engineering stress-strain curves of the SSR-T6 and IM-T6 samples tested along longitudinal and transversal directions of the tube, respectively. The corresponding average tensile mechanical properties of the tested specimens are shown in Table 2. It can be seen that the average yield strength (YS), ultimate tensile strength (UTS), and elongation to fracture of the SSR(L)-T6 and SSR(T)-T6 samples were 262 ± 0 MPa, 276 ± 3 MPa, and 9.0 ± 0.8%, as well as 276 ± 7 MPa, 296 ± 7 MPa, and 7.6 ± 1.1%, respectively, illustrating a slight anisotropy of the tensile mechanical properties in the two directions. On the other hand, the average YS, UTS, and elongation to fracture of the IM(L)-T6 and IM(T)-T6 samples were 305 ± 4 MPa, 321 ± 3 MPa, and 9.8 ± 0.5%, as well as 304 ± 5 MPa, 320 ± 5 MPa, and 8.6 ± 1.2%, respectively, suggesting that tensile mechanical properties of the IM-T6 samples were almost isotropic. All the tensile property values reach the GB/T 3191-2019 standard.

Figure 7 shows the SEM secondary electron images of transversal fracture surfaces of the T6 heat treated SSR and IM samples along the longitudinal and transversal directions, respectively. It can be seen that the fracture surfaces of the SSR(L)-T6 and SSR(T)-T6 samples exhibited large amounts of cavities and dimples with sizes in the range of 5–20 μm, which may associate with the oxide particles with a size range of 1–5 μm (Figure 7a,c). It can be envisaged that the cavities might result from the fracture of oxide particles and the decohesion between oxide particles and matrix, and the dimples were the result of ductile fracture of the metal matrix. On the other hand, the fracture surfaces of the IM(L)-T6 and IM(T)-T6 samples showed a large number of dimples with sizes of 5–20 μm, which likely originated from plastic deformation during tensile testing, as shown in Figure 7b,d.

As shown in Figure 8a,c, the longitudinal fracture of the SSR(L)-T6 and SSR(T)-T6 samples contained a large number of cavities with sizes of 2–10 μm located between the oxide particles. This is probably due to the debonding between oxide particles and the Al matrix during the tensile test. However, for the fractured IM(L)-T6 and IM(T)-T6 samples, the number density of the cavities was much lower, and their sizes were also smaller, being 1–5 μm, as shown in Figure 8b,d.

## 4. Discussion

### 4.1. Microstructure Evolution

This study shows that a microstructure consisting of coarse elongated grains and fine equiaxed grains formed in the as-extruded SSR sample. The formation of the coarse elongated grains is likely due to dynamic recrystallization and following rapid grain growth induced by severe plastic deformation during hot extrusion. In contrast, the presence of fine equiaxed grains may associate with the pinning effect of oxide particles on grain boundaries which inhibited further growth of some recrystallized grains. It is noted that this microstructure is quite different from that of the IM sample produced by the extrusion of a cast ingot under the same condition. The latter demonstrates a homogeneous microstructure consisting of nearly full recrystallized equiaxed grains with the lack of oxide particles in the Al matrix, suggesting that the fine oxide particles have a significant influence on the microstructure evolution of the SSR sample.

It is believed that the fine oxide particles likely originated from the breaking of the Al_2_O_3_ layers on the chips since the significant difference in Young’s modulus between Al and Al_2_O_3_ (Al: 68 GPa, Al_2_O_3_: 300 GPa), which causes stress concentration on the Al_2_O_3_ layers during hot pressing and hot extrusion. The high shear stress produced by plastic deformation during extrusion facilitates the breaking of the Al_2_O_3_ layers. In addition, it is believed that Mg atoms segregate from grain interiors to the inter-chip boundaries during extrusion at elevated temperatures, and the segregated Mg could react with Al_2_O_3_ layers through the following reactions due to the more negative free energy of formation of MgO than that of Al_2_O_3_ [37,38,39]:3Mg + 4Al_2_O_3_ = 3MgAl_2_O_4_ + 2Al(1)
3Mg + Al_2_O_3_ = 3MgO +2Al(2)

These reactions could also facilitate the breaking of the Al_2_O_3_ surface films by introducing shear stress as a result of volume change during the reactions [40]. This finding is supported by the EDS mapping results of the oxide particles showing that the Mg element is detected in the oxide particles, as shown in Figure 4b. Similar oxide particles were also observed in the microstructures of samples produced by hot extrusion of compacts of AZ91 magnesium alloy scraps [19] and 5083 aluminum alloy chips [13] due to the breaking of oxide films.

It is well recognized that the coarsening of the recrystallized grains usually occurs at high temperatures due to grain growth, which is confirmed by the EBSD results of the IM sample, demonstrating a significant increase of Al grain size from 35 to 60 μm after T6 heat treatment, as shown in Figure 2b,d. In contrast, for the SSR sample, an apparent grain refinement occurs, as evidenced by the decrease of the average grain size from 60 to 34 μm after T6 heat treatment (Figure 2a,c). This decrease in the average grain size is likely due to further recrystallization caused by residual strain formed in the SSR sample. It is believed that amounts of dislocations are generated during plastic deformation associated with extrusion, and these dislocations are accumulated due to the pinning effect of the fine oxide particles on dislocations. Moreover, the strain incompatibility between oxide particles and matrix will also cause the formation of dislocations at the particles/matrix interfaces [41]. As a result, these dislocations are kept in a matrix as a result of the quenching of the extruded tube. Hence, it can be speculated that the accumulation of dislocations in the vicinity of oxide particles leads to the stress concentration around them, contributing to the formation of residual strain which provides the thermodynamic driving force for the recrystallization during the solution treatment stage of the T6 heat treatment. This is strongly supported by the observation of the increase in the fraction of HAGB from 53% in the SSR tube sample to 80% in the SSR(L)-T6 sample, as shown in Figure 2a,c.

It is reported [17] that the fracture of oxide layers contributes to a quick establishment of atomic bonding between individual chips. Once the breaking of oxide surface layers occurs, rapid atomic diffusion and high-quality metallurgical bonding can be achieved. However, most of the current studies showed that atomic diffusion bonding across the inter-chip boundaries was weakened due to the insufficient breaking of the oxide surface layers [25,26]. In the present study, due to the severe plastic deformation caused by a high extrusion ratio of 58:1, sufficient breaking of the oxide surface layers occurs during extrusion. This facilitates rapid atomic diffusion and the establishment of high-quality inter-chip bonding, contributing to the good tensile mechanical properties of the recycled materials. It can be envisaged that the fine oxide particles distributed along inter-chip boundaries, which inhibited grain boundaries diffusion, and grain growth occurred only along the inter-chip boundaries, resulting in the formation of elongated grains. This was strongly supported by the observation of the presence of elongated coarse grains in the as-extruded and T6 heat treated SSR tube samples, as shown in Figure 2a,c,e. In contrast, due to the lack of the inhibition effect of oxide particles on grain growth, almost all Al grains in the microstructure of the as-extruded and T6 heat treated IM samples are equiaxed, as shown in Figure 2b,d,f.

### 4.2. Mechanical Properties

It is noted that, after T6 heat treatment, the SSR samples showed nearly similar ductility with that of the IM tube samples, as reflected by the average elongation to fracture values of 9.0% and 9.8% in the samples tested along the longitudinal direction, and that of 7.6% and 8.6% in the samples tested along the transversal direction. This indicates that the inter-chip boundaries with a high level of atomic bonding are established in the SSR sample through the hot pressing of the chips followed by hot extrusion. Meanwhile, the fine oxide particles distributed along the inter-chip boundaries do not cause the ductility of the recycled material to deteriorate significantly, which is probably due to their small sizes of less than 2 μm. Since the oxide particles are so small, their spacing is large enough that the cavities between the oxide particles and matrix are unlikely to connect to each other for further crack nucleation and growth, which contributes to the good ductility of the solid-state recycled materials. This conjecture is supported by the SEM observations of the longitudinal sections of the fracture surfaces in the tensile test specimens of the SSR(L)-T6 and SSR(T)-T6 samples (Figure 7a,c). Similar results have been reported in solid-state recycled Al-7%Si-0.3%Mg alloy and AZ91 Mg alloy, which demonstrates that the ductility of the recycled materials are not affected by the small silicon particles [17] and oxide particles [19] with sizes of less than 10 μm.

It is interesting to find that the tensile strengths of the T6 heat treated SSR samples tested along the transversal direction of the tube are better than those of the samples tested along the longitudinal direction of the tube (Table 2). This demonstrates that a better tensile strength is achieved in the T6 heat treated SSR sample along the direction normal to the inter-chip boundaries, which confirms that a high-quality inter-chip bonding is established between the chips. Based on the same tensile strengths of IM(L)-T6 and IM(T)-T6, the influence of straightening deformation on transverse specimens can be excluded. A slightly lower ductility of transversal specimens is likely due to the different specimen dimensions. Considering that the SSR(L)-T6 and SSR(T)-T6 samples have similar average grain sizes of 34 and 42 μm respectively, the influence of grain size difference on the strength can be negligible. Therefore, it is speculated that the slight increase of tensile strength along the transversal direction of the tube may associate with the higher density of grain boundaries produced due to the presence of elongated grains distributed along the direction parallel to inter-chip boundaries.

It is noted that the T6 heat treated SSR sample exhibited a lower tensile strength than that of the T6 heat treated IM sample (Table 2 and Figure 6). This can be attributed to the lower density of β″ precipitates associated with a lower Mg content caused by the consumption of Mg atoms by the oxidation reaction during hot pressing and hot extrusion. It is supported by the chemical compositions of samples after T6 heat treatment that Mg element in SSR(L)-T6 was consumed, causing a lower Mg content compared with IM(L)-T6 (see Table 3). Ahmed et al. [38] also reported a similar reaction between Mg and Al_2_O_3_ surface layers of Al7075 powder particles in Al7075/SiC composites prepared by powder metallurgy routine, and the consumption of some of the Mg element in the matrix, drastically reducing the number density of MgZn_2_ precipitates in this material.

## 5. Conclusions

In the present work, a solid-state recycled AA6061 tube was successfully produced by hot pressing of chips to produce a chip compact, followed by hot extrusion of the chip compact. The hot extrusion resulted in the formation of fine oxide particles due to the breaking down of oxide layers on the chips and the reaction between Mg and Al_2_O_3_. Dynamic recrystallization of the Al grains and establishment of high-level inter-chip atomic bonding also occurred during the chip consolidation process. The fine oxide particles inhibited further grain growth of the recrystallized grains in the particle-rich zones, leading to the formation of a microstructure including coarse elongated grains with sizes of 420 μm, and fine equiaxed grains with sizes of 10 μm. T6 heat treatment caused an apparent Al grain refinement of the recycled tube sample with grain sizes of 34 μm, which was likely attributed to further recrystallization induced by residual strains around the oxide particles. The good tensile ductility of 7.6–9% in the T6 heat treated solid-state recycled tube sample was possibly related to the high-level inter-chip bonding and the fine oxide particles which were small enough that further crack nucleation and growth around them were inhibited during a tensile test. Compared to the ingot metallurgy sample, the lower tensile strength 276 MPa of the T6 heat treated solid-state recycled tube sample might be associated with the consumption of Mg atoms by the oxidation reaction, leading to the lower density of β″ precipitates in precipitation strengthening.

## Figures and Tables

**Figure 1 materials-16-01384-f001:**
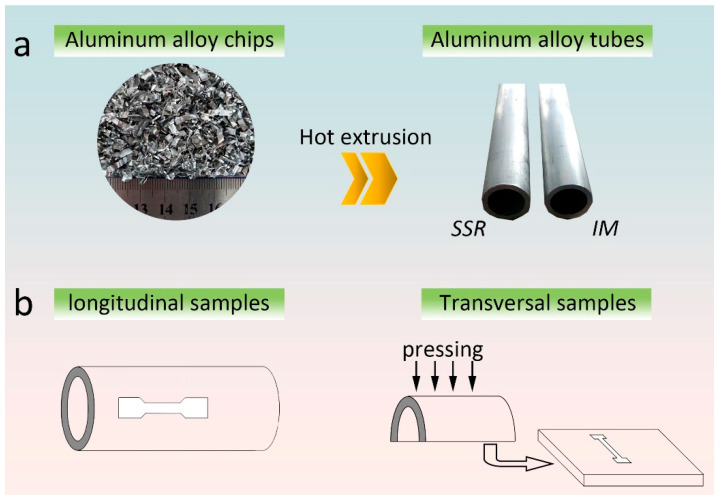
(**a**) As-received AA6061 aluminum alloy chips and as-extruded SSR and IM tube samples; (**b**) Schematic drawings showing the positions of the tensile test specimens cut along the longitudinal and transversal directions of the tube samples.

**Figure 2 materials-16-01384-f002:**
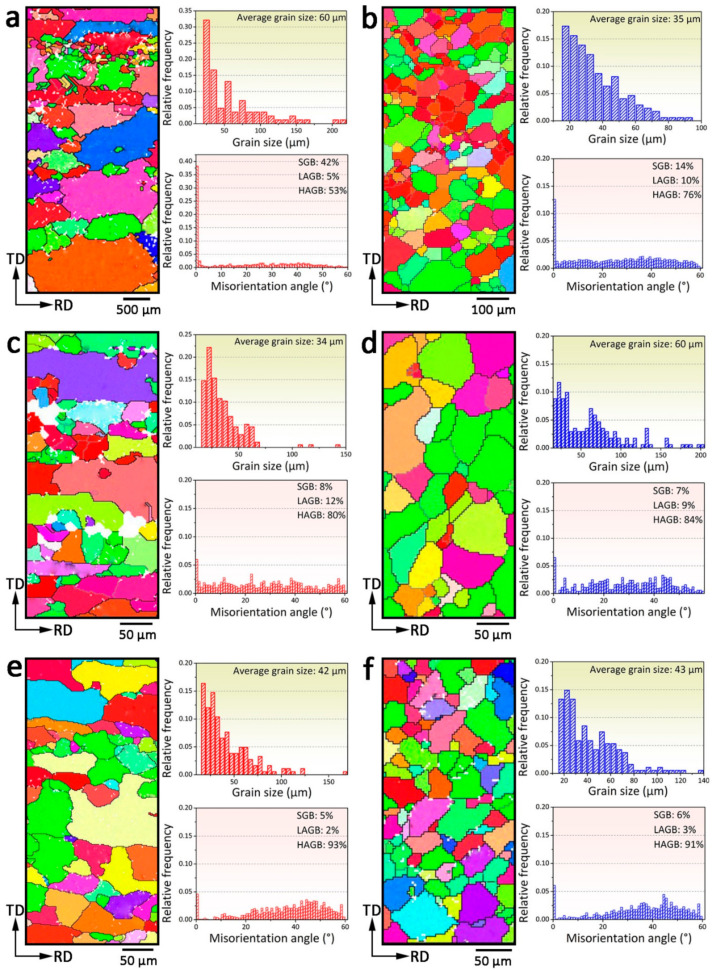
EBSD IPF, grain size distributions, and misorientation angle distributions of (**a**) SSR, (**b**) IM, (**c**) SSR-T6, (**d**) IM-T6, (**e**) SSR(T)-T6, and (**f**) IM(T)-T6 samples.

**Figure 3 materials-16-01384-f003:**
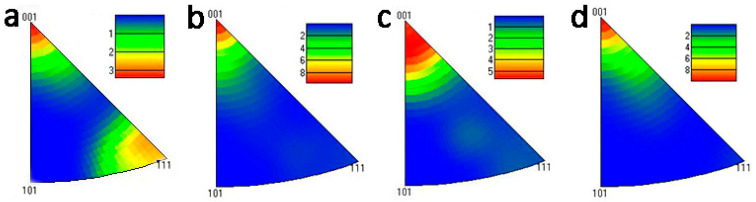
Inverse pole figures of the as-extruded and T6 heat treated samples: (**a**) SSR, (**b**) IM, (**c**) SSR-T6, and (**d**) IM-T6.

**Figure 4 materials-16-01384-f004:**
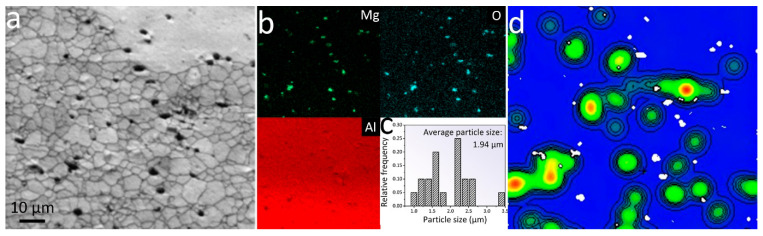
(**a**) EBSD band contrast map, (**b**) EDS mapping, (**c**) particle size distribution, and (**d**) strain contouring image of fine equiaxed grain zone of the SSR sample.

**Figure 5 materials-16-01384-f005:**
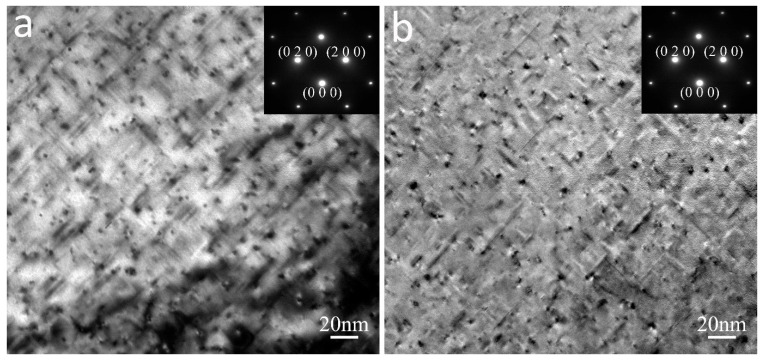
TEM bright field images and corresponding SAED patterns of (**a**) SSR-T6 and (**b**) IM-T6 samples.

**Figure 6 materials-16-01384-f006:**
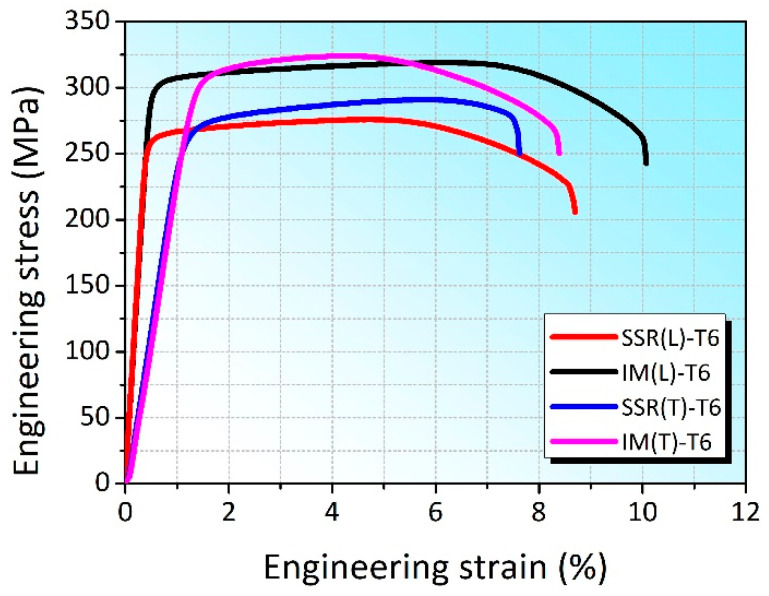
Engineering stress-strain curves of the SSR-T6 and IM-T6 samples tested along both longitudinal and transversal directions of the tubes.

**Figure 7 materials-16-01384-f007:**
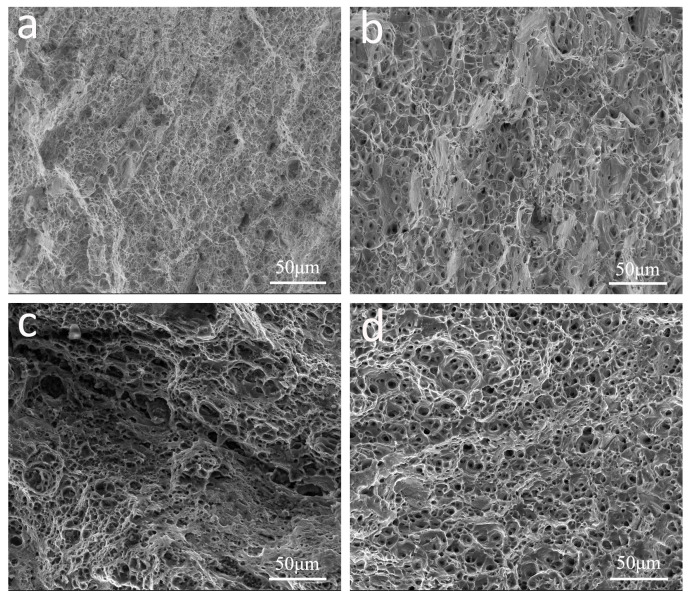
SEM secondary electron images of fracture surfaces of the tensile test specimens: (**a**) SSR(L)-T6, (**b**) IM (L)-T6, (**c**) SSR(T)-T6, and (**d**) IM(T)-T6.

**Figure 8 materials-16-01384-f008:**
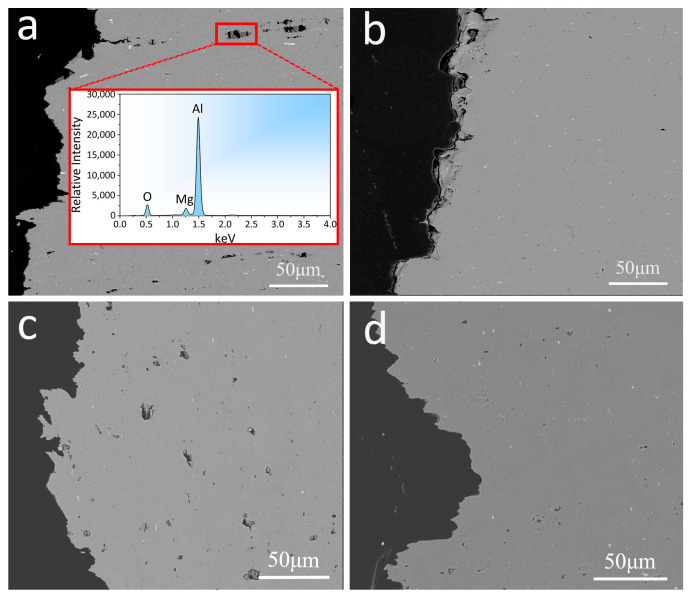
SEM backscatter election images of longitudinal sections below the fracture surfaces of the tensile specimens: (**a**) SSR(L)-T6, (**b**) IM (L)-T6, (**c**) SSR(T)-T6, and (**d**) IM(T)-T6.

**Table 1 materials-16-01384-t001:** Sample notations and descriptions of aluminum alloy tubes.

Notations	Sample Descriptions
SSR	Al chips + hot extrusion
SSR(L)-T6	Al chips + hot extrusion + T6 heat treatment, cut along the longitudinal direction
SSR(T)-T6	Al chips + hot extrusion + hydraulic press + T6 heat treatment, cut along the transversal direction
IM	Al cast ingot + hot extrusion
IM(L)-T6	Al cast ingot + hot extrusion + T6 heat treatment, cut along the longitudinal direction
IM(T)-T6	Al cast ingot + hot extrusion + hydraulic press + T6 heat treatment, cut along the transversal direction

**Table 2 materials-16-01384-t002:** Tensile mechanical properties of the SSR-T6 and IM-T6 samples along the longitudinal (L) and transversal (T) directions (±: standard deviation of the mean).

Samples	YS (MPa)	UTS (MPa)	Elongation to Fracture (%)
SSR(L)-T6	262 ± 0	276 ± 3	9.0 ± 0.8
IM(L)-T6	305 ± 4	321 ± 3	9.8 ± 0.5
SSR(T)-T6	276 ± 7	296 ± 7	7.6 ± 1.1
IM(T)-T6	304 ± 5	320 ± 5	8.6 ± 1.2
GB/T 3191-2019	240	260	8

**Table 3 materials-16-01384-t003:** Chemical compositions of the SSR and IM tubes after T6 heat treatment.

Samples	Elemental Content (wt.%)				
	Mg	Si	Fe	Others	Al
SSR(L)-T6	0.77	0.53	0.13	<0.1	Bal.
IM(L)-T6	0.95	0.59	0.12	<0.1	Bal.

## Data Availability

All data are provided in full in the results section of this paper.

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
