# Peer review of "Effects of Oxide Fragments on Microstructure and Mechanical Properties of AA6061 Aluminum Alloy Tube Fabricated by Thermomechanical Consolidation of Machining Chips"

_materials, 2023, doi:10.3390/ma16041384_

Round 1
Reviewer 1 Report
Dear authors,
Thanks for sharing your work on solid state aluminum recycling. The field is of high interest since the EV deployment and the Green Deal policies around the world are pushing research for less energy demanding extruded products. The approach to the problem is novel and the contribution is relevant regarding the explanation of the mechanisms behind the control of microstructure and mechanical properties.
General comments: The introduction is well structured and back-ups the importance of the field and shows relevant previous work on the approaches from other research groups. The experimental technique and material description is easy to understand and follow but lacks enough information for reproducibility in laboratories working in the same field.
The results section is well ordered and explained. Minor information missing for exact interpretation of some results should be added.
The discussion section is properly ordered and explained, easy to follow.
Conclusions sohuld consider some effects of experimental result dispersion and measurement uncertainty so that they are stronger backed-up.
Specific Comments:
1. Chip characterization doesn't allow the reproducibility of the experimental setup. Please address type of machining, actual cutting parameters and employed lubricant. If not possible, at least apparent and tap densities should be indicated for a rough orientation.
2. As oxidation stands for a major factor in the paper, oxygen measurements should be reported, if available, in each step of the process. This would allow addressing the source of the oxidation and orient further research for its handling in favour of the reciclability via solid state.
3. The point above should be combined with the actual chemical composition of the ingot and the chips, so that the dicussion about the effect of available Mg depletion by oxidation is undoubtedly supported. Please include a table the actual chemical composition measured in the materials under study, at least in terms of Fe, Mg and Si contents.
4. The effect of cleaning the chips and the convenience of employing alternative cryogenic lubrication should be at least mentioned, as cleaning itself should be accounted in life cycle assessment.
5. The hot pressing conditions are not decribed enough and they are of mayor relevance for potential replication by third party researchers. Type of material for the tooling, heating time and method and final ingot density should be mentioned.
6. 6061 is whenever possible quenched and aged without an intermediate solubilization, like the one used in the paper. The thickness of the extruded tube allows direct quenching and subsequent ageing. It should at least be mentioned the rationale behind the chosen extrusion temperature and holding time and the second solubilization.
7. Machining tensile specimens with WEDM usually leaves crack initiators along the burned surface. Were the samples grinded clean from any white layer for safety?
8. Gauge length strongly affects elongation measurements and they are extremelly small compared to usual standard lengths for extruded materials. A note should be included about this effect and the extensometry method should also be mentioned.
9. In Figure 4.a. Was it meant to say "BSE" instead of "EBSD"?
10. In Figure 4.b. The size distribution graph is hard to read. Please improve the readability/resolution.
11. In Table 2. Please specify if the plus/minus refers to the standard deviation of the individual results or an estimation of standard deviation of the mean. A table or annex with the individual testing results would also clarify this point.
12. Value dispersion is extremely low compared to regular ISO 17025 lab results. Please address load cell uncertainty, extensometer and section measurement device uncertainty (calipper or micrometer) from the calibration certificates of the equipments. Or at least mention load cell and extensometer grade and reference standard for the readers to assess the result dispersion.
12. According the grain size distribution graphs, stating that the shape is bimodal could be open to statistical discussion. It would be enough to change the reference to "bimodal" by a reference to "abnormal grain growth" to get clear of this point if authors would prefer to avoid the issue.
13. Does the straightening deformation on transverse specimens influence their very good behavior? This may contribute to the differences between "L" and "T" samples. Mentioning it in the paper would cover a point discussion.
To sum up and finish the review, I would like to congratulate the authors for this very interesting work. Thanks for sharing with the community.
Author Response
Dear Reviewer,
I am pleased to resubmit the revised version of Materials-2177036 entitled with “Effects of oxide fragments on microstructure and mechanical properties of AA6061 aluminum alloy tube fabricated by thermomechanical consolidation of machining chips”. Thank you for all the comments. We do appreciate your constructive suggestions which are valuable for improving the quality of our manuscript. As requested, we have addressed each concern from you and outlined these below.
Comments:
Point 1:
Chip characterization doesn't allow the reproducibility of the experimental setup. Please address type of machining, actual cutting parameters and employed lubricant. If not possible, at least apparent and tap densities should be indicated for a rough orientation.
Response 1:
Thanks for your suggestion. The aluminum alloy chips are as-received from industry, the machining, cutting parameters and lubricant are unknown. We have measured the tap density of the chips, that is 0.33 g/cm3. Also, we have added the corresponding description in Experimental Procedure section. Please see Page 4 (highlight in yellow).
Point 2:
As oxidation stands for a major factor in the paper, oxygen measurements should be reported, if available, in each step of the process. This would allow addressing the source of the oxidation and orient further research for its handling in favor of the recyclability via solid state.
Response 2:
Thanks for your suggestion. In this paper, we stated that the source of the oxidization was from the oxide layers on the surface of chips. The oxide layers were broken during the hot pressing which were conducted under protection of argon and led to a high density of 99.5%. This could prevent further oxidation of the chips during the following hot extrusion and heat treatment. Therefore, it is believed that the source of the oxidation is mainly deriving from the surface oxide layers of the original chips.
The oxygen contents of SSR and IM tube are measured by a LECO TCH-600 nitrogen/oxygen/hydrogen analyzer, which are 2.52 wt.% and 0.45 wt.%. Please see the corresponding data in Page 5 (highlight in yellow).
Point 3:
The point above should be combined with the actual chemical composition of the ingot and the chips, so that the discussion about the effect of available Mg depletion by oxidation is undoubtedly supported. Please include a table the actual chemical composition measured in the materials under study, at least in terms of Fe, Mg and Si contents.
Response 3:
Thanks for your constructive suggestion. We determined the chemical compositions of the SSR and IM tube sample after T6 heat treatment by an inductive coupled plasma emission spectrometer. It shows that the Mg content of SSR is lower than that of IM which has the same chemical composition with the original chips, suggesting that the Mg depletion by oxidation is undoubtedly supported. The corresponding results are listed in Table 3. Please see Page 12 (highlight in yellow).
Point 4:
The effect of cleaning the chips and the convenience of employing alternative cryogenic lubrication should be at least mentioned, as cleaning itself should be accounted in life cycle assessment.
Response 4:
Thanks for your suggestion. The main purpose of this study is to investigate the effects of oxide fragments on the microstructure evolution and mechanical properties. The chips were cleaned in order to promote the metallurgical bonding between the individual chips as well as remove the impurities. Therefore, the effects of cleaning and lubrication were ignored in this study.
Point 5:
The hot pressing conditions are not described enough and they are of mayor relevance for potential replication by third party researchers. Type of material for the tooling, heating time and method and final ingot density should be mentioned.
Response 5:
Thanks for your suggestion. The more detailed hot pressing conditions are described as: “The dried chips were dropped into a cylindrical H13 steel die and then heated by ceramic heater band up to 400°C (holding time: 10 min) under the protection of argon, followed by hot pressing (200 ton) for 5 min.” Please see Page 4 (highlight in yellow).
Point 6:
6061 is whenever possible quenched and aged without an intermediate solubilization, like the one used in the paper. The thickness of the extruded tube allows direct quenching and subsequent ageing. It should at least be mentioned the rationale behind the chosen extrusion temperature and holding time and the second solubilization.
Response 6:
Thanks for your question. In this study, in order to obtain the microstructure of extruded state, the tube was quenched in water after hot extrusion against the occurrence of static recrystallization for a short time after hot extrusion. After hot extrusion, the tube samples were performed standard T6 heat treatment including solution treatment, quenching and artificial aging. Therefore, the effects of oxide particles on microstructure evolution during hot extrusion and solution treatment were well studied. Many attempts were done to explore the extrusion temperature, a fine-grained microstructure could be obtained under 480 ℃ which was finally chosen as the extrusion temperature.
Point 7:
Machining tensile specimens with WEDM usually leaves crack initiators along the burned surface. Were the samples grinded clean from any white layer for safety?
Response 7:
Thanks for your question. Yes, the samples were ground to 2000 mesh abrasive paper. Please see Page 6 (highlight in yellow).
Point 8:
Gauge length strongly affects elongation measurements and they are extremely small compared to usual standard lengths for extruded materials. A note should be included about this effect and the extensometry method should also be mentioned.
Response 8:
We added the description in the tensile test section that “specimen size was not following the standard due to the limited dimension of samples” and “extensometer was used to measure the elongation to fracture”. Please see Page 6 (highlight in yellow and blue).
Point 9:
In Figure 4.a. Was it meant to say "BSE" instead of "EBSD"?
Response 9:
It should be EBSD. The Fig. 4a was detected by EBSD and band contrast map was addressed by Channel 5.
Point 10:
In Figure 4.b. The size distribution graph is hard to read. Please improve the readability/resolution.
Response 10:
Thanks for your suggestion. The image has been changed to a new one with high resolution.
Point 11:
In Table 2. Please specify if the plus/minus refers to the standard deviation of the individual results or an estimation of standard deviation of the mean. A table or annex with the individual testing results would also clarify this point.
Response 11:
The plus/minus refers to the standard deviation of the mean. We have added the relative description in the table caption. Please see Page 11 (highlight in yellow).
Point 12:
Value dispersion is extremely low compared to regular ISO 17025 lab results. Please address load cell uncertainty, extensometer and section measurement device uncertainty (calipper or micrometer) from the calibration certificates of the equipments. Or at least mention load cell and extensometer grade and reference standard for the readers to assess the result dispersion.
Response 12:
The tensile test was performed based on GB/T 228-2002 standards (excluding specimen size due to the limited dimension of samples). Extensometer was used to measure the elongation to fracture. The load cell and extensometer are Grade 1. We added it in Page 6 (highlight in blue).
Point 13:
According the grain size distribution graphs, stating that the shape is bimodal could be open to statistical discussion. It would be enough to change the reference to "bimodal" by a reference to "abnormal grain growth" to get clear of this point if authors would prefer to avoid the issue.
Response 13:
Thanks for your suggestion. I have deleted mentions to “bimodal”.
Point 14:
Does the straightening deformation on transverse specimens influence their very good behavior? This may contribute to the differences between "L" and "T" samples. Mentioning it in the paper would cover a point discussion.
Response 14:
Thanks for your question. Although the transverse specimens were subjected to the straightening deformation, the specimens were also annealed during the following solution treatment at high temperature. It can be found that the T6 heat treated IM tube samples along the transversal and longitudinal direction showed the same tensile strength. A slight lower ductility of transverse specimens is likely due to the different specimen dimensions. In light of it, it can be speculated that the straightening deformation on transverse specimens does not influence the tensile behavior. We added a point discussion “Based on the same tensile strengths of IM(L)-T6 and IM(T)-T6, the influence of straightening deformation on transverse specimens can be excluded. A slight lower ductility of transverse specimens is likely due to the different specimen dimensions”. on Page 16 (highlight in yellow).
Reviewer 2 Report
- Some comments are provided below that could be helpful for the authors to improve their manuscript.
1) English (language, sentence structure and grammar) needs improvement.
2) Abstract should contain part of your Methodology and investigation outcomes.
3) Keywords should be differ from the paper title.
4) Remove “Oxide particles” from Keywords.
5) The work evaluated requires a thorough review by the authors.
6) The introduction should show that the proposed objectives are new or contribute new knowledge to the topic under study. Additionally, it should be increased to a maximum of two pages.
7) Please, add the references after “Experimental Procedure” section.
8) Where we can find the SEM, “The microstructures of the tube samples were characterized by scanning electron microscopy (SEM)? Page 3, Line 96-97.
9) Could you explain what the colors means in EBSD IPF analysis???
10) No tolerance for Tensile mechanical properties measurement, its fixed value (mean result).
11) Conclusion part should be one paragraph.
12) Add references from 2018 till date.
Good Luck
Author Response
Dear Reviewer,
I am pleased to resubmit the revised version of Materials-2177036 entitled with “Effects of oxide fragments on microstructure and mechanical properties of AA6061 aluminum alloy tube fabricated by thermomechanical consolidation of machining chips”. Thank you for all the comments. We do appreciate your constructive suggestions which are valuable for improving the quality of our manuscript. As requested, we have addressed each concern from you and outlined these below.
Comments:
Point 1:
English (language, sentence structure and grammar) needs improvement.
Response 1:
Thanks for your suggestion. The English has been checked and improved.
Point 2:
Abstract should contain part of your Methodology and investigation outcomes.
Response 2:
Thanks for your suggestion. The abstract has been revised somewhere to mention the methodology and outcomes. Please see Page 1-2 (highlight in green).
Point 3:
Keywords should be differ from the paper title.
Response 3:
Thanks for your suggestion. The keywords have been revised. Please see Page 2 (highlight in green).
Point 4:
Remove “Oxide particles” from Keywords.
Response 4:
The “Oxide particles” in Keywords has been removed.
Point 5:
The work evaluated requires a thorough review by the authors.
Response 5:
Yes, the work has been reviewed thoroughly again by us.
Point 6:
The introduction should show that the proposed objectives are new or contribute new knowledge to the topic under study. Additionally, it should be increased to a maximum of two pages.
Response 6:
Thanks for your significant suggestion. The introduction section has been revised to show clear objective and contribution. In addition, the length is increased to almost two pages. Please see Page 4 (highlight in green).
Point 7:
Please, add the references after “Experimental Procedure” section.
Response 7:
The references are listed in the “Reference” section.
Point 8:
Where we can find the SEM, “The microstructures of the tube samples were characterized by scanning electron microscopy (SEM)? Page 3, Line 96-97.
Response 8:
The microstructure of the fracture surfaces of tensile test specimens were characterized by SEM. Please see Fig. 7 and 8.
Point 9:
Could you explain what the colors means in EBSD IPF analysis???
Response 9:
The colors represent the grain orientation, different color means the grains have different orientations. Please see color code figure in the below.
Fig. R1 EBSD color code.
Point 10:
No tolerance for Tensile mechanical properties measurement, its fixed value (mean result)
Response 10:
The standard deviations of the tensile mechanical property values were shown in Table 2. Now we also have added the standard deviations in the text. Please see Page 11 (highlight in green).
Point 11:
Conclusion part should be one paragraph.
Response 11:
Thanks for your suggestion. The conclusion has been revised to be in one paragraph. Please see Page 17 (highlight in green).
Point 12:
Add references from 2018 till date.
Response 12:
Thanks for your significant suggestion. We have added some new references from 2018 till date.

Reviewer 3 Report
This will review the “Effects of oxide fragments on microstructure and mechanical properties of AA6061 aluminum alloy tube fabricated by thermomechanical consolidation of machining chips” manuscript. The manuscript is well-written and contains interesting results related to the extrusion of 6061 alloy under different thermal conditions. However, the following points need to be considered by the authors.
1- In lines 119 to 137, you have only mentioned the changes in grain size due to heat treatment. This is despite the fact that you mentioned heat treatment as the only reason for grain size changes. This is true. But it is not enough. You should provide a stronger analysis.
2- In line 148, what is the basis of the authors for the occurrence of recrystallization? At what temperature did recrystallization occur?
3- In line 155, the results of the EDS analysis should be expressed as a possible phase (referring to Figure 4).
4- Since the axis of strengthening of 6xxx alloy in this research is based on β" phase, it is necessary to identify this phase with XRD.
Author Response
Dear Reviewer,
I am pleased to resubmit the revised version of Materials-2177036 entitled with “Effects of oxide fragments on microstructure and mechanical properties of AA6061 aluminum alloy tube fabricated by thermomechanical consolidation of machining chips”. Thank you for all the comments. We do appreciate your constructive suggestions which are valuable for improving the quality of our manuscript. As requested, we have addressed each concern from you and outlined these below.
Comments:
Point 1:
In line 119 to 137, you have only mentioned the changes in grain size due to heat treatment. This is despite the fact that you mentioned heat treatment as the only reason for grain size changes. This is true. But it is not enough. You should provide a stronger analysis.
Response 1:
Thanks for your question. In line 119 to 137, we compared the microstructure, mainly the grain size, between the extruded state and the heat treated state samples. It is undoubted that the changes in grain size was owing to heat treatment because only heat treatment was performed after hot extrusion. In the discussion section, we also mentioned that the reason of changes in grain size. For the SSR tube, the decrease of average grain size after heat treatment is likely due to further recrystallization caused by residual strain formed in the as-extruded SSR sample. The accumulation of dislocations in the vicinity of oxide particles leads to the stress concentration around them, contributing to the formation of residual strain which provides the thermodynamic driving force for the recrystallization during solution treatment stage of the T6 heat treatment. For the IM tube, the increase of grain size can be explained by grain growth. Please see the discussion in Page 14 (highlight in grey).
Point 2:
In line 148, what is the basis of the authors for the occurrence of recrystallization? At what temperature did recrystallization occur?
Response 2:
Thanks for your question. According to the inverse pole figure in Fig. 3, it can be observed that SSR tube had very strong <100> and weak <111> fiber texture, SSR(L)-T6 changed a lot that <100> fiber texture was much stronger, however, <111> fiber texture was weaker and even disappeared. For FCC metals with high stacking fault energy (SFE) like aluminum, the deformation texture of them after uniaxial extrusion is described as a double fiber texture with major <111> deformation texture and small amounts of <100> recrystallization texture. The recrystallization texture of aluminum consists of a strong cube texture {001}<100>. The inverse pole figures results confirm that large amounts of deformed structures in SSR tube were remained after hot extrusion and they were restored in SSR-T6 via recrystallization. In general, the recrystallization temperature Tr = (0.35~0.45)Tm, where Tm is melting temperature. The solution treatment temperature of T6 heat treatment was 535 ℃, which is enough for the occurrence of recrystallization. We added the discussion in Page 7 (highlight in grey).
Point 3:
In line 155, the results of the EDS analysis should be expressed as a possible phase (referring to Figure 4).
Response 3:
The result of the EDS shows that the particles were in rich of Al, Mg and O elements. The possible phase would be MgAl2O4, which has been reported in literatures. We mentioned it the discussion section. We added the descriptions as “It is speculated that the particle is likely MgAl2O4 and will be mentioned in the discussion section”. Please see Page 10 (highlight in grey).
Point 4:
Since the axis of strengthening of 6xxx alloy in this research is based on β" phase, it is necessary to identify this phase with XRD.
Response 4:
Thank you very much for the suggestion. It is really necessary to identify the β" phases which have important influence on mechanical properties of 6xxx Al alloy. However, due to the Chinese Spring Festival holiday, all the labs are closed and the XRD experiment is not carried out. As a matter of fact, in order to identify the β" phases we have tried to do the selective-area electron diffraction analyses on TEM (please see Fig. 5), but unfortunately the diffraction spots of β" phases were not detected due to the weak signal. The further characterization of the β" phases will be done by HRTEM in our future work.
Reviewer 4 Report
Journal: Materials (ISSN 1996-1944)
Manuscript ID: materials-2177036
The authors presented a paper on “The effects of oxide fragments on the microstructure and mechanical properties of AA6061 aluminum alloy tube produced by thermomechanical consolidation of machining chips”. Recusing waste chips with recycling is very important in terms of sustainability. I think the article is well organized and suitable for the "Materials" journal. However, the article will be ready for publication after a major revision. Comments are listed below.
1. Turnitin similarity rate is 27%.
2. Hot pressing is not mentioned in the introduction. It should be explained by giving examples from the literature. In addition, the introduction section should be expanded.
3. In the last paragraph of the introduction, the original aspect of the study should be mentioned and its difference from other studies should be explained.
4. The authors used argon gas during hot pressing in "Experimental Procedure" (page 2, line 82). Why was argon gas chosen? Why weren't other protective atmosphere gases (H, N etc.) used?
5. The chemical composition of AA6061 aluminum alloy should be shown.
6. 400 oC According to which parameters was the sintering temperature determined?
7. Pressing pressure and time are not given. It should be added.
8. According to which standards was the tensile test performed?
9. Little is known about tensile test results. It should be supported by the literature.
10. The relationship between the development of microstructure and mechanical properties was not mentioned. It should be explained.
11. The article contains numerous typographic and language errors. It should be corrected.
12. The article should be rearranged by taking into account the journal writing rules and citation rules.
13. The paper is well-organized, yet there is a reference problem. First, your reference list contains no article from the “Materials” journal. If your work is convenient for this journal's context, then there are many references from this journal. Secondly, cited sources should be primary ones. Namely, the indexed area shows the power of a paper and directly your paper's reliability. Please make regulations in this direction.
*** Authors must consider them properly before submitting the revised manuscript. A point-by-point reply is required when the revised files are submitted.
Author Response
Dear Reviewer,
I am pleased to resubmit the revised version of Materials-2177036 entitled with “Effects of oxide fragments on microstructure and mechanical properties of AA6061 aluminum alloy tube fabricated by thermomechanical consolidation of machining chips”. Thank you for all the comments. We do appreciate your constructive suggestions which are valuable for improving the quality of our manuscript. As requested, we have addressed each concern from you and outlined these below.
Comments:
Point 1:
Turnitin similarity rate is 27%.
Response 1:
Thanks for your significant suggestion. We also check the similarity of the manuscript and have revised some sentences where is similar to the literatures. We believe that the similarity has been reduced.
Point 2:
Hot pressing is not mentioned in the introduction. It should be explained by giving examples from the literature. In addition, the introduction section should be expanded.
Response 2:
Thanks for your suggestion. Now, we have added one examples from the study of Yang et al., which demonstrated that sufficient bonding was established between individual swarf pieces of Ti-6Al-4V during the short hot pressing. Please see Page 2-3 (highlight in blue). In addition, we response the point 3 and other comments on introduction section from Reviewers. The introduction section has been expanded.
Point 3:
In the last paragraph of the introduction, the original aspect of the study should be mentioned and its difference from other studies should be explained.
Response 3:
Thanks for your constructive suggestion. The previous works in literatures mainly reported that a high quality inter-chip bonding could be established in the solid state recycling of aluminum chips, as the breaking of oxide layers enhanced the atomic diffusion across the inter-chip boundaries. However, the effects of oxide fragments on microstructure evolution and mechanical properties are still unknown. This paper mainly focuses on characterizing the oxide particles and investigating the rationale behind the anisotropic microstructure as well as tensile mechanical properties in different directions by the pinning effects of oxide particles, which has never been studied. The last paragraph of the introduction is re-organized to show the differences and contributions from other studies. Please see Page 4 (highlight in green).
Point 4:
The authors used argon gas during hot pressing in "Experimental Procedure" (page 2, line 82). Why was argon gas chosen? Why weren't other protective atmosphere gases (H, N etc.) used?.
Response 4:
Thanks for your question. Because the aluminum chips are in small scale, the surface area are very high. In order to prevent oxidation of aluminum chips so that good metallurgical bonding can be achieved, inert gas should be used as the protective atmosphere. H2 is an active gas, which is dangerous at high temperature. N2 is an inert gas, but it is prone to react with Al to form AlN [1]. Argon is a desirable inert gas to prevent oxidation of aluminum chips without a reaction with Al. In light of it, argon gas was chosen in this study.
[1] Schaffer, G., and B. Hall. "The influence of the atmosphere on the sintering of aluminum." Metallurgical and Materials Transactions A 33.10 (2002): 3279-3284.
Point 5:
The chemical composition of AA6061 aluminum alloy should be shown.
Response 5:
Thanks for your suggestion. The chemical composition is measured by ICP and the result is listed in Table 3. Please see Page 12.
Point 6:
400 ℃ According to which parameters was the sintering temperature determined?
Response 6:
Thanks for your question. The aluminum chips are recycled in a solid state way. The process temperature should be lower than the solid state temperature (550 ℃ for AA6061). In addition, the material of the hot pressing die is H13 steel, and the safe working temperature range of H13 steel should be lower than 450 ℃. 400 ℃ is enough high to reach a good metallurgical bonding, which led to that the density of hot pressed compacts was as high as 99.5%.
Point 7:
Pressing pressure and time are not given. It should be added.
Response 7:
Thanks for your suggestion. The pressing pressure was 200 ton and holding time was 5 min. We have added related descriptions in Page 4 (highlight in yellow).
Point 8:
According to which standards was the tensile test performed?
Response 8:
Thanks for your question. The tensile test was performed based on GB/T 228-2002 standards (excluding specimen size due to the limited dimension of samples). We add it in Page 6 (highlight in Blue).
Point 9:
Little is known about tensile test results. It should be supported by the literature.
Response 9:
Thanks for your suggestion. We checked the standard of tensile properties of 6061-T6 aluminum alloy. All the samples reach the standard tensile properties. We added the standard values in the Table 2. Please see Page 11 (highlight in blue).
Point 10:
The relationship between the development of microstructure and mechanical properties was not mentioned. It should be explained.
Response 10:
In general, to explain the relationship of microstructure and mechanical properties, the analysis of strengthening mechanisms is usually conducted. In this study, it is believed that grain boundary strengthening, dispersed particles strengthening and precipitation strengthening are the main strengthening mechanisms to dominate the mechanical properties. However, it is difficult to gather statistics of volume fraction and size of particles and β" precipitates, and a deep TEM characterization is needed to demonstrate the coherent relationship of β" precipitates and Al matrix. It is an interesting direction to explore and we will study this point in the future work.
In this study, we mainly focus on the anisotropic microstructure of the solid state recycled samples correlated to their tensile properties of longitudinal and transverse specimens. Consequently, we speculate that the slight increase of tensile strength along transversal direction of the tube may associate with the higher density of grain boundaries produced, which is likely due to the presence of elongated grains distributed along the direction parallel to inter-chip boundaries.
Point 11:
The article contains numerous typographic and language errors. It should be corrected.
Response 11:
Thanks for your suggestion. We have checked the manuscript carefully. Some typographic and language errors have been corrected.
Point 12:
The article should be rearranged by taking into account the journal writing rules and citation rules.
Response 12:
Thanks for your suggestion. We have rearranged somewhere including writing rules and citation following the rules in instructions of the journal.
Point 13:
The paper is well-organized, yet there is a reference problem. First, your reference list contains no article from the “Materials” journal. If your work is convenient for this journal's context, then there are many references from this journal. Secondly, cited sources should be primary ones. Namely, the indexed area shows the power of a paper and directly your paper's reliability. Please make regulations in this direction.
Response 13:
Thanks for your suggestion. We searched some references published in “Materials” relevant to solid state recycling of aluminum alloy chips and cited some of them in the corresponding position of this manuscript.
Round 2
Reviewer 1 Report
Dear authors, Thanks for the answers.
Looking forward to see future works from your lab.
Regards
Reviewer 2 Report
Dear Author (s)
Good work.,,
I have two additional comments:
1) Add a paragraph in the introduction about your work objectives and the gap covered by the curent paper.
2) The conclusion should be a single paragraph that does not separate points.
Good luck
Reviewer 3 Report
The manuscript is improved and recommended for publication
Reviewer 4 Report
Journal: Materials (ISSN 1996-1944)
Manuscript ID: materials-2177036
Review Report R2#
The authors completed all requested corrections. In my opinion, this article can be accepted for publication in the "Materials" journal in its final form.